# The Role of Different Earthworm Species (*Metaphire Hilgendorfi* and *Eisenia Fetida*) on $CO_2$ Emissions and Microbial Biomass during Barley Decomposition

**Toru Hamamoto [1],*** and **Yoshitaka Uchida [2]**

1    Graduate School of Agriculture, Hokkaido University, Sapporo Hokkaido 060-8589, Japan
2    Research Faculty of Agriculture, Hokkaido University, Sapporo Hokkaido 060-8589, Japan;
     uchiday@chem.agr.hokudai.ac.jp
*    Correspondence: hamatoru@chem.agr.hokudai.ac.jp; Tel.: +81-11-706-2405

**Abstract:** Earthworms are commonly known as essential modifiers of soil carbon (C) and nitrogen (N) cycles, but the effects of their species on nutrient cycles and interaction with soil microbial activities during the decomposition of organic materials remain unclear. We conducted an incubation experiment to investigate the effect of two different epigeic earthworms (*M. hilgendorfi* and *E. fetida*) on C and N concentrations and related enzyme activities in agricultural soils with added barley residues (ground barley powder). To achieve this, four treatments were included; (1) *M. hilgendorfi* and barley, (2) *E. fetida* and barley, (3) barley without earthworms, and (4) without earthworms and without barley. After 32 days incubation, we measured soil pH, inorganic N, microbial biomass C (MBC), water or hot-water soluble C, and soil enzyme activities. We also measured $CO_2$ emissions during the incubation. Our results indicated the earthworm activity in soils had no effect on the cumulative $CO_2$ emissions. However, *M. hilgendorfi* had a potential to accumulate MBC (2.9 g kg$^{-1}$ soil) and nitrate-N (39 mg kg$^{-1}$ soil), compared to *E. fetida* (2.5 g kg$^{-1}$ soil and 14 mg kg$^{-1}$ soil, respectively). In conclusion, the interaction between soil microbes and earthworm is influenced by earthworm species, consequently influencing the soil C and N dynamics.

**Keywords:** epigeic earthworm; soil carbon; soil nitrogen; soil enzyme activity; agricultural soil

## 1. Introduction

Organic amendment can be used to increase soil carbon (C) and nitrogen (N), which are the indicators of agricultural production [1,2]. The degradation of organic matter and physical disturbance of soils, performed by the earthworms, can alter soil microbial community, and thus influence rates of the C and N cycles following the organic amendment [3–6]. Barley is one of the major crop species in the world and it is important to fully understand how the C from barley cycles in agricultural soils. Thus, a study will be required to understand interactions among earthworm and soil C dynamics, particularly using barley as organic materials.

Differences in earthworm species composition affect microbial biomass and N retention in soils because of their different feeding strategies [7–10]. Epigeic earthworms are litter dwellers and inhabit the surface level of soils. Thus, these earthworms enhance decomposition rates of their habitats interacting with microorganisms and other organisms [8–10]. Better understanding of the impact of epigeic earthworms on C and N dynamics in agricultural soils will provide basic information to evaluate the efficacy of organic amendments to maintain soil health.



Uchida et al. [11] reported that *Metaphire hilgendorfi*, which is one of the dominant species in Hokkaido, Japan, ingests litter and soil faster and more efficiently compared to other epigeic species. This species was also reported to decrease the biomass of both gram-positive and gram-negative bacteria in soils [12]. In contrast, Enami et al. [13] investigated a microbial response by the presence of *M. hilgendorfi*, and found that the gram-negative bacteria increased in the soils. Another earthworm type, *Eisenia fetida* (one of the major epigeic species), had no effect on water-soluble C in soils [14], while Zhang et al. [15] and Aira et al. [16] reported that *E. fetida* have the potential to improve C mineralisation and increase microbial activity related to decomposition. These contradictory results might be due to the lack of information about the interaction among earthworms, soil C dynamics and soil microbial activities.

To assess the changes in soil microbes, the measurements of soil microbial enzyme activities are commonly used, regarding the decomposition of organic matter and nutrient cycles [17]. Those enzymatic activities, which are particularly related to N dynamics, are strongly correlated with the addition of organic C [18,19]. However, studies observing the interaction between the earthworm species and soil N enzyme activities are few [17,20,21].

Therefore, we conducted an incubation study and our aims were to investigate the role of the two earthworm species on soil C and N dynamics, and microbial changes with organic materials application. We hypothesized that the earthworm species change the patterns of soil C and N with the presence of organic matter: *M. Hilgendorfi* increases soil microbial biomass and thus have more potential to immobilize C to soils, while *E. fetida* promotes soil C mineralisation.

## 2. Materials and Methods

### 2.1. Soil Characteristics and Earthworms

We conducted an incubation experiment using an agricultural soil, plant residues (finely ground barley (*Hordeum vulgare*) grain) from a private company (ITOEN Ltd., Tokyo, Japan) and two different earthworm species. The soil (0–5 cm depth) was collected from the experimental maize (*Zea mays*) farm of Hokkaido University, Sapporo, Japan (43°04′31.6″N, 141°20′03.4″E, 13 m above sea level) in August 2017. At the sampling, the soil moisture was 68.9 ± 0.6% as gravimetric and the bulk density was 0.48 ± 0.03 g cm$^{-3}$ (n = 5). Soils were air-dried and sieved to 2 mm. The total C and N in soils were 42.3 ± 1.2 g kg$^{-1}$ soil and 3.60 ± 0.29 g kg$^{-1}$ soil, respectively (*n* = 3). C and N concentrations of the barley were 44.9 ± 0.3% and 2.0 ± 0.2%, respectively (*n* = 3). Total C and N content of the samples were determined using an elemental analyser (2400 Series; PerkinElmer Inc., MA., USA). Two types of earthworms were used; one was *M. hilgendorfi* sampled from the forest in Hokkaido University in August 2017 and the other was *E. fetida* obtained from a private company (Iwase farm, Ibaraki, Japan).

### 2.2. Experimental Design

The soils were packed into 2 L glass bottles (12.2 cm diameter, 5 cm depth and 573 g soil bottle$^{-1}$). The bulk density of the soil in the bottles was 1.0 g cm$^{-3}$. Soil moisture (gravimetric) was maintained at 30% (bottle weight basis) during the incubation period, which is optimum soil moisture content for an earthworm species *Lumbricus terrestris* [22]. The weight of each bottle was measured every day and distilled water was added to replace the evaporated water. Before the application of the barley and the earthworms, the bottles with soils were pre-incubated for one week to allow passing the bacterial flush induced by drying–rewetting of soils and/or soil physical disturbance by sieving [23–26].

Then, four earthworms (one of the two species) and the barley grains (5 g) were added to each bottle (equivalent of 342 individuals m$^{-2}$). Before the earthworms were added to the bottles, the feed remaining within each earthworm was emptied by placing the earthworms on an agar plate with water (>12 h). This was performed to minimize the addition of C to soils through earthworms. Immediately before the application, the average living mass of *M. hilgendorfi* and *E. fetida* was 6.70 ± 0.18 and 1.14 ± 0.04 g per bottle (573 and 96 g m$^{-2}$), respectively. The earthworm density was determined in

accordance to the review by Fründ et al. [27], showing that the typical earthworm density (all species concerned) in temperate region was 10–1000 individuals $m^{-2}$. Vidal et al. [26] determined 25 mg of dry matter $g^{-1}$ fresh weight $day^{-1}$ as a good compromise to favour earthworm survival. Thus, we applied the same quantity; 5 g of barley (2.25 g C $kg^{-1}$ soil) was added to each bottle. The treatments were (1) *M. hilgendorfi* and barley; M, (2) *E. fetida* and barley; E, (3) barley without earthworms; B, and (4) without earthworms and without barley (Control); C, with three replicates. Those four treatments were incubated for 32 days to investigate soil C dynamics when the earthworms are actively decomposing the organic materials. The length of the incubation period was determined since there were no significant differences in $CO_2$ emissions after day 20 followed by the commencement of the incubation.

### 2.3. Measurements of Soil Chemical Properties

After the 32 days incubation, the soils were sampled from the surface (0–2 cm depth) excluding barley residue and mixed for further analysis. For the determination of the inorganic-N concentrations (ammonium-N; $NH_4^+$-N, nitrate-N; $NO_3^-$-N), 5 g samples were extracted with 25 mL of 10% KCl. After 30 min shaking, the extractant was filtered through filter paper. Then, inorganic-N concentrations in soil samples were measured using a colorimetric method with a flow injection analyser (AQLA-700; Aqualab, Tokyo, Japan) as described previously [28]. The water-soluble C (WSC) and hot water-soluble C (HWSC) were measured using a modified previously reported method [29]. Sub-samples of 3 g each were placed in centrifuge bottles and shaken with 30 mL of distilled water at 20 °C for 1 h. The samples were then centrifuged for 30 min at 4000 rpm, the supernatant was filtered through filter paper (Grade 5C, <5 mm; Advantec, Tokyo, Japan) and the amount of soluble organic C was determined using a total C analyser (TOC 5000A; Shimadzu, Japan). This was the WSC fraction of the soil organic C. A further 30 mL of distilled water was added to the sediments remaining in the centrifuge tubes after the supernatants (WSC) were decanted and the tubes were capped and left for 16 h in a hot-water bath at 80 °C. The samples were then centrifuged, filtered, and analysed for soluble organic C concentrations using the same methods as for the WSC analysis. This was the HWSC fraction of the soil organic C.

The microbial biomass C (MBC) was measured using a modified chloroform fumigation method of Vance et al. [30] and Toda and Uchida [30,31]. Two samples of 2 g each were taken from each soil; one was immediately extracted with 10 mL of 0.5 M $K_2SO_4$ and filtered through filter paper, whereas the other 2 g samples were fumigated under chloroform for 48 h using an evacuated desiccator and then extracted in the same way. The extracts were stored in a freezer until C analysis. At that point, extracts were diluted at 1:10 with water and organic C was determined in 25 mL aliquots of the diluted samples using a total organic C analyser (TOC-5000A; Shimadzu, Kyoto, Japan). The amount of organic C in the non-fumigated samples was subtracted from the amount of organic C in the fumigated samples; then, values were multiplied by 2.64 to convert them to MBC.

### 2.4. Measurements of Soil Enzyme Activities

Urease activity (UA) and nitrification enzyme activity (NEA) were measured using modified previously reported methods [32–34]. For UA, 5 g soil sample taken from each bottle was mixed with 10 mL of 0.08 mol $L^{-1}$ aqueous urea solution. After incubation for 2 h, 50 mL solution (1M KCl and 0.01M HCl) was added, and the mixtures shaken for 30 min. The resulting suspensions were filtered, and the filtrates analysed as described above in the soil N measurement section. For NEA, a 5 g soil sample taken from each bottle was placed in a 50 mL polyethylene bottle and incubated for 4 days at room temperature (25 °C) after the addition of 0.2 mg N $g^{-1}$ soil of ammonium sulphate. $NO_3^-$-N concentrations were measured calorimetrically as explained above in the soil N measurement section. NEA was estimated based on the difference between the $NO_3^-$-N concentration in the fresh soils before the commencement of the incubation and $NO_3^-$-N concentration in the soils after the 4 days incubation. Denitrification enzyme activity (DEA) was estimated by measuring the increases in $N_2O$ concentrations during a soil incubation using an acetylene block technique. The method was modified

based on Mogi et al. [34]. A 3 g soil sample taken from each bottle was incubated in a 100 mL vial with 3 mL of a solution with excess amounts of C and N substrates (2 g C $L^{-1}$ as glucose and 200 mg N $L^{-1}$ as $KNO_3$). The air in the headspace was completely replaced by $N_2$ gas, and then 10% of the $N_2$ gas was replaced with acetylene. After 2 h incubation, 30 mL of headspace gas was sampled. Sampled $N_2O$ gases were measured by gas chromatograph (GC-2014; Shimadzu Co., Japan).

Substrate-induced respiration rate (SIR) and specific growth rate (SGR) were measured for each of the soil samples, the equivalent of 50 g dry soil in glass bottles (0.9 L). Each soil sample was amended with glucose at a rate of 1 mg C $g^{-1}$ soil. After amendment, each glass bottle was sealed by lid with tubes, which allowed air to pass from outside to the gas analyser (Isotopic $CO_2$ Analyser; Los Gatos Research, CA, USA) with a Multi-port Inlet Unit (MIU). To minimize the effect of any contamination (e.g., human respiration), the ambient air was continuously pumped from outside using two air pumps during analysis (non-noise W1000; Japan Pet Design, Tokyo, Japan). Excess ambient air allowed to pass outside to avoid the accumulation of respired $CO_2$ in the bottles. The air flow was controlled by a purge-meter to avoid the accumulation of respired $CO_2$ in the bottles (P-710; Tokyo Keiso, Tokyo, Japan). The flow rate of inflowing air was set at approximately 50 mL $min^{-1}$, and the outflowing air was regulated as 35 mL $min^{-1}$. Each bottle was measured for five minutes and switched to the next bottle by MIU after measurement. Thus, $CO_2$ emission from each bottle was measured every 70 min (12 samples and 2 times of ambient air) and continuously analysed up to 15 h. SIR was calculated by average of $CO_2$ emission rate for 3.5 to 4.6 h after glucose amendment. The SIR phase was followed by an exponential growth phase (with a specific growth rate, SGR), which continued until the substrate availability in the soil water solution becomes limiting. SGR was calculated by linear regression after logarithmic transformation of the $CO_2$ emission rate.

### 2.5. Measurements of Soil Respiration

Soil respiration rates were monitored throughout the incubation period for 32 days and measured using a nondispersive infrared gas analysis sensor (C12329-01; Hamamatsu Photonics K.K., Shizuoka, Japan) equipped with an air-pump (EAP-01; AS ONE, Osaka, Japan). Each incubation bottle was sealed by lid with two tubes (inlet and outlet), which allowed air to circulate in the bottle headspace and the gas analysis sensor. Silicon rubber was used to seal the gap within the circulation loop. The soil respiration concentration was calculated based on the increase of $CO_2$ concentrations in the bottle headspace over 30 min.

### 2.6. Data Analysis

The data was analysed using one-way analysis of variance (ANOVA), while the time course of $CO_2$ emission was analysed using a mixed model for repeated measurements. Tukey's test was performed for the analysis of significant differences for each treatment. $CO_2$ emissions were log-transformed ($log_{10}$(flux)), to achieve normal distribution, prior to analysis. To compare the effects of $CO_2$ emission on earthworm and barley application, control treatment value was excluded from other treatments. After that, the data were analysed using a mixed model for repeated measurements to investigate earthworm and barley amendments. All statistical analyses were performed in R ver. 3.4.1 with a threshold *P*-value of 0.05.

## 3. Results

Barley and earthworm application significantly influenced soil chemical properties compared to soils without barley (Control). Soil pH in water was increased by barley application, while such an increase was not observed when barley was applied with *M. hilgendorfi* (Figure 1a). At the same timing, the amount of $NO_3^-$-N in the E and B treatments were significantly lower than that in the M and Control treatment (Figure 1b). The WSC concentration in soils was increased by the addition of the barley, particularly in E and B treatment (Figure 1c). There was no effect of B treatment on MBC compared to Control treatment (Figure 1d). Contrastingly, M treatment resulted in the highest MBC

concentrations ($2.9 \pm 0.2$ g kg$^{-1}$ soil), whereas E treatment showed the lowest MBC concentrations ($2.5 \pm 0.1$ g kg$^{-1}$ soil). For SGR, there was a significantly lower growth rate when applied barley (Figure 1e).

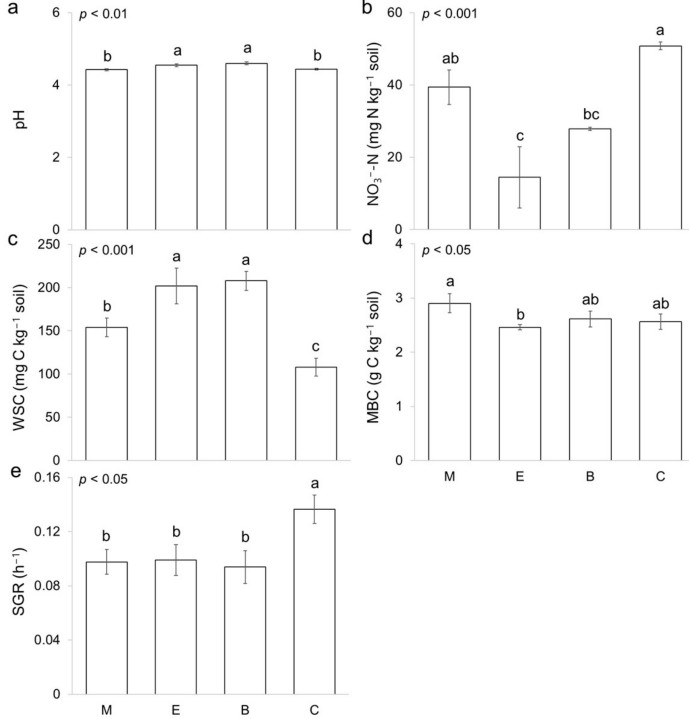

**Figure 1.** Soil pH (**a**), amount of soil nitrate (NO$_3$$^-$-N, **b**), water soluble carbon (WSC, **c**), microbial biomass carbon (MBC, **d**), and specific growth rate (SGR, **e**) under different treatments. The treatments include *M. hilgendorfi* and barley; M, *E. fetida* and barley; E, barley without earthworms; B, and without earthworms and without barley (Control); C. Level of significance was determined by one-way analysis of variance (ANOVA). Different letters indicate significant differences between treatments ($p < 0.05$). Error bars represent one standard deviation ($n = 3$).

There were no significant differences among the treatments regarding soil enzyme activities (e.g., UA, NEA and DEA) and N and C related characteristics (Table 1). The cumulative soil CO$_2$ emissions were significantly higher with barley ($p < 0.05$), compared to without barley (Figure 2). Regarding the time course changes in soil CO$_2$ emissions, no significant differences among barley application were seen (Figure 3). The soil respiration of barley-only treatment was lower at the beginning (day 0), and CO$_2$ emissions rate increased and peaked at day 8 and decreased towards the end of incubation.

**Table 1.** The non-significant ($p > 0.05$) effects of earthworms on soils.

| | **M** | **E** | **B** | **C** |
|---|---|---|---|---|
| NH$_4$$^+$-N (mg N kg$^{-1}$ soil) | $24.1 \pm 10.6$ | $14.7 \pm 9.2$ | $21.0 \pm 2.9$ | $20.0 \pm 1.7$ |
| TN (g N kg$^{-1}$ soil) | $5.43 \pm 0.25$ | $5.28 \pm 0.37$ | $4.52 \pm 1.29$ | $5.58 \pm 0.26$ |
| HWSC (mg C kg$^{-1}$ soil) | $815 \pm 24$ | $867 \pm 123$ | $800 \pm 8$ | $843 \pm 68$ |
| TC (g C kg$^{-1}$ soil) | $45.8 \pm 6.29$ | $39.6 \pm 0.37$ | $39.4 \pm 0.2$ | $40.7 \pm 0.08$ |
| CN ratio | $8.49 \pm 1.54$ | $7.53 \pm 0.45$ | $9.69 \pm 3.39$ | $7.31 \pm 0.34$ |
| UA (mg NH$_4$$^+$-N kg$^{-1}$ soil h$^{-1}$) | $7.61 \pm 6.82$ | $23.5 \pm 4.04$ | $19.5 \pm 10.11$ | $17.5 \pm 9.31$ |
| NEA ($\mu$g NO$_3$$^-$-N kg$^{-1}$ soil h$^{-1}$) | $194 \pm 42$ | $233 \pm 59$ | $91 \pm 58$ | $121 \pm 41$ |
| DEA ($\mu$g N$_2$O-N kg$^{-1}$ soil h$^{-1}$) | $158 \pm 97$ | $85 \pm 7$ | $93 \pm 18$ | $66 \pm 16$ |
| SIR (mg CO$_2$ kg$^{-1}$ soil h$^{-1}$) | $3.34 \pm 0.73$ | $3.14 \pm 0.44$ | $3.50 \pm 0.60$ | $2.15 \pm 0.15$ |

Note: Level of significance was determined by one-way ANOVA. The treatments include *M. hilgendorfi* and barley; M, *E. fetida* and barley; E, barley without earthworms; B, and without earthworms and without barley (Control); C. Error values represent standard deviations ($n = 3$).

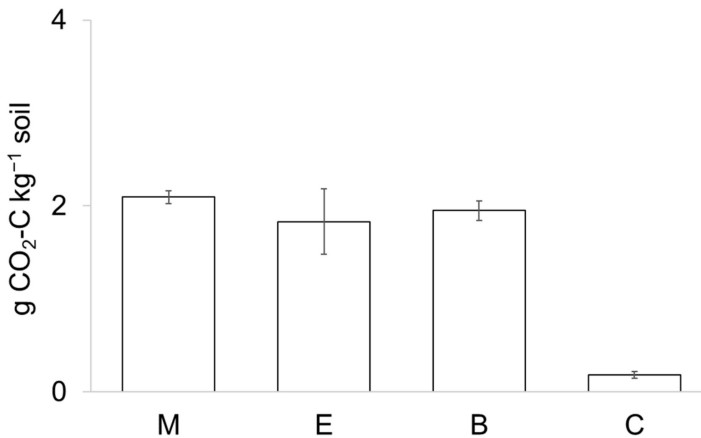

**Figure 2.** Cumulative amount of $CO_2$-C evolved from soils throughout the experiment. The treatments include *M. hilgendorfi* and barley; M, *E. fetida* and barley; E, barley without earthworms; B, and without earthworms and without barley (Control); C. Error bars represent one standard deviation ($n = 3$).

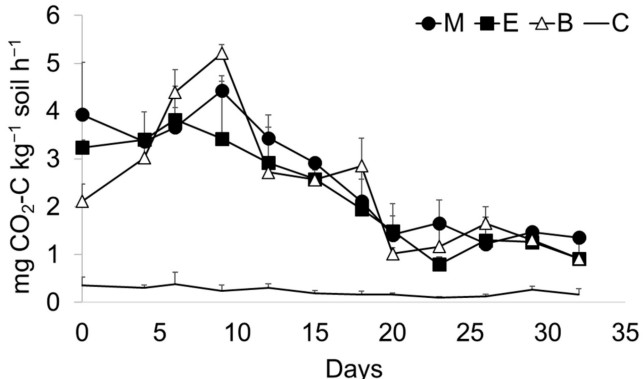

**Figure 3.** The time course for $CO_2$ emissions throughout the experiment. Circles represent the soils with barley and *M. hilgendorfi* (M) whereas squares represent the soils with barley and *E. fetida* (E). Triangle and line-only depict soils with and without barley (B and C), respectively. Error bar represents one standard deviation ($n = 3$).

## 4. Discussion

Organic amendments increased the WSC concentration, compared to the soils without barley treatments. Bastida et al. [35] reported that the increase of WSC could be due to mineralisation processes. In addition, a previous study reported that soil pH was increased by barley application because of enhanced C mineralisation [36]. However, in our study, *M. hilgendorfi* suppressed the effect of barley application on the increase in WSC and soil pH when compared to *E. fetida*. Contrastingly, for MBC, the *M. hilgendorfi* treatment showed the highest values within all the treatments. These facts suggest that with the presence of *M. hilgendorfi*, C derived from barley tends to be immobilized and remained in soils as MBC. Thus, *M. hilgendorfi* had a positive impact regarding the increase in soil microbes after the addition of organic materials compared to the soils with barley and the soils with barley and *E. fetida*. Similarly, Enami et al. [13] observed a significant increase in amounts of total and bacterial phospholipid-derived fatty acids due to the presence of *M. hilgendorfi* with rice straw. In contrast, a previous study reported that *E. fetida* decreased MBC [37]. Ruz-Jerez et al. [38] suggested that *E. fetida* might become the competitors against microbes for carbonaceous substrates as energy and this might be a reason for the decreased MBC. Another study also suggested that *Lumbricus* species (same family as *E. fetida*) selectively consume soil fractions with a high concentration of microbes [39]. According to a previous study, those two earthworms are distributed in similar agricultural ecosystems and in a similar density [40]. We also note that the earthworm distribution depends on agricultural

management systems (e.g., mulch), and the application of organic materials often positively impacts the activity of earthworms. Thus, the information regarding these differences between the earthworm species can be used to establish strategies to efficiently increase soil C using agricultural residues.

*M. hilgendorfi* increased the amount of $NO_3^--N$ in soils in this study, compared to the soils with *E. fetida*. To support this, Kawaguchi et al. [41] observed that the cast of *M. hilgendorfi* became hotspots of nitrification. We also visually observed more holes within the soils with *M. hilgendorfi* compared to *E. fetida*, although we did not measure the physical changes in the soils during the incubation. The aeration of the soils (physical changes in the soils) could be a reason for the stimulation of N mineralisation and nitrification. Greiner et al. [42] also found that *M. hilgendorfi* improved the soil aggregate formation, resulting in N mineralisation. In contrast to *M. hilgendorfi*, our studies showed a lower $NO_3^--N$ concentration the soil with barley and *E. fetida*. This result may suggest the gaseous loss of $NO_3^--N$ during organic matter decomposition. Also, there is a possibility of the acceleration of N assimilation by *E. fetida*, although further studies are needed to confirm this [43,44].

Overall, the earthworms influenced the substrate availability and microbial biomass, but not the other measured "potentials" of microbial enzyme activities related to N cycles (e.g., nitrification and denitrification potentials). One of the reasons for this might be because our study was conducted only for 32 days. A longer incubation period might accumulate more casts by earthworms and further influence the potentials of soil microbial activities. *M. hilgendorfi* may have a long-term effect of microbial activities because the amount of those casts, which become a major source of soil aggregates formation, will reach more than 20 t ha$^{-1}$ year$^{-1}$ [11,41]. Additionally, the activity of earthworm creates heterogeneous conditions in the soils because of the mucus secreted from the earthworm and casting activity [45]. The soil near the holes created by earthworm activities often show improved biological activity, thus stimulated soil microbes, compared to bulk soils [46]. Because our study was an incubation study with a relatively smaller amount of soil, we sampled soils and then homogenised for each analysis. Thus, this might have diluted the impacts of the earthworm compared to the control soils, particularly enzyme activities.

Our incubation study suggested that even when we applied the same type of organic matter into soils, its impacts on soil microbes differ depending on the earthworm species. Thus, further study will be required to understand interaction with earthworms and nutrient cycles at the field condition (e.g., earthworm populations) [47].

The current study also indicated that the earthworm activity in soils had no effect on the cumulative $CO_2$ emissions for 32 days. A previous quantitative review reported that the earthworm presence increased soil $CO_2$ emissions by 33% on average, but the results also stated that the data from the short-term (< 30 days) studies were markedly variable [48,49]. Also, we did not observe clear differences in $CO_2$ emissions between samples with and without earthworms. This might depend on the amount of soil C. A similar study using *E. fetida* showed no significant effect on soil $CO_2$ emissions from soils (total C = 1.71%) [50]. On the other hand, Caravaca et al. [14] reported that $CO_2$ emissions were increased by earthworms in soils with depleted C condition (total C = 0.32%). The $CO_2$ emissions were positively related to the amount of C in soils, in general [51]. Thus, $CO_2$ emissions derived from earthworms might be negligible particularly in soils with higher C contents. In our study, the C contents of the soil was 4.23%. Thus, in our soils, soil microbes were active both with and without the presence of earthworm, actively emitting $CO_2$.

## 5. Conclusions

This study reports that two different epigeic earthworms (*M. hilgendorfi* and *E. fetida*) with the presence of barley residue differently influenced soil C and N dynamics. The soils with *M. hilgendorfi* and barley accumulated MBC and $NO_3^--N$ more efficiently compared to the soils with *E. fetida* and barley. In contrast, no significant differences were observed between the soils with different earthworms regarding the soil enzyme activities. Future research on the effect of earthworm on the C and N cycle, then consequent microbial community change under different soil C contents, is recommended to fully

understand soil C accumulation in the field scale. Particularly, the microbial mechanisms to sequester C in soils (e.g., cell wall production) and their interaction with the presence of earthworms will be an interesting topic to be investigated.

**Author Contributions:** Conceptualization T.H. and Y.U.; methodology, T.H.; investigation, T.H.; data analysis, T.H.; writing—original draft preparation, T.H.; writing—review and editing, Y.U.; visualization, T.H.

**Acknowledgments:** The authors thank Mr. Taira at Field Science Centre for Northern Biosphere, Hokkaido University, Japan, for technical support.

**Conflicts of Interest:** The authors declare no conflict of interest.

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
