# Peer review of "The Role of Different Earthworm Species (Metaphire Hilgendorfi and Eisenia Fetida) on CO2 Emissions and Microbial Biomass during Barley Decomposition"

_sustainability, doi:10.3390/su11236544_

Round 1
Reviewer 1 Report
Line 18, suggested "barely without earthworms" to cause less confusion.
Line 40, probably adding a sentence describing why these two species are important (so readers understand why authors chosen these two for this study).
Line 46, I searched a little and found some paper about comapring metaphire guillelmi vs. eisenia fetida (https://www.sciencedirect.com/science/article/pii/S0038071700001115). would it be helpful to add some papers in Intro or Discussion reporting the similar species but not exactly the same?
In the Intro, probably be nice to talk about the importance of barely? otherwise, readers might wonder why using barely but not other types of OM.
Line 65, were these measurements from this study or from other published sources? add reference
Line 90, how long was this incubation? please add days. and why 32 days (based on Intro)?
Line 91, how authors sampled this 5 g materials? was it sampled before mixing the whole thing? or only from the surface?
Line 149,was soil moisture monitored over time along with soil respiration?
Line 158, in figure 1 I saw you have different letters for significant differences, did authors use Tukey HSD ? please clarify it in the Data analysis
For Results, please put p values following the sentences about stats, and please do not just put p value < 0.05. put the exact p values received from the stats unless it just said < 0.001
Line 213, add several sentences describing what the normal distribution of earthworm species would be in the real field. and are those invasive earthworms?
Line 232-235, same amount of OM (mass) might not contain the same amount of OC and nutrients
Line 238, were they conducted in the same range of incubation period? 32 days? and the same background material (e.i. barley)?
Line 245, were there any reasons to explain the differences? authors in the Discussion only described the difference among studies but did not really go deep to explain (at least try) why.
Line 252, this sentence was too general, authors could give more specific suggestions for future directions of research.
Reviewer 2 Report
Manuscript ID: sustainability-631758
Title: The Role of Different Earthworm Species (Metaphire Hilgendorfi and Eisenia Fetida) on Carbon Dynamics during Barley Decomposition
The paper is publishable, but only after revision and I did have a few comments that the authors should consider as they revise the paper. I ask that the authors specifically address each of my comments in their response.
Major Comments:
1.The reviewer believes that authors should determine the place of two (MAIN! (L47)) Earthworm Species in the earthworms community, taking into account soil diversity. This is especially true for arable soils. In general, the soil part, or rather its absence, is a weak link in this article. It is impossible in experience to be limited to the name SOIL. The reader will have questions. If barley decomposition is studied, then it can take place only in arable soils. At the same time, the authors write on L67 that the Metaphire Hilgendorfi species was obtained from the forest, and according to L63, the Reviewer assumes that it was from the forest litter, and on L27 (Keywords), it was stated that the study was agricultural soil. Thus, the authors must prove that their choice of two species was not accidental and it is determined by the dominance of these species in agricultural soils. For example, the species Lumbricus terrestris (L72) was mentioned several times, but why did the authors not study it?
2.How was the experiment period lasting 32 days justified? This is even more relevant question, since the authors enter into a discussion with themselves (L237-238) about the representativeness of such a duration.
3.The name includes Carbon Dynamics, which implies a change in time. This is not consistent with the results. The experiment did not provide such data. The dynamic aspect is reflected in Figure 3. The time course for CO2 emissions ... But this is not Carbon Dynamics!
4.Section 4 is designed in such a way that it is very difficult to single out the personal achievements of the authors. A lot is involved in the discussion of other opinions and links.
5.Our Journal allows colour drawings (for example, for Fig. 3 this would be very appropriate). But even if the authors chose a black and white style, then Fig. 1 and 2 do not look aesthetically pleasing. Moreover, most importantly, the black colour does not allow to see the lower limit of the limit. Make the fill a rare hatch or speck.
6.Author Contributions. In the presence of two co-authors, it is logical to show the contribution of each, since it is, clear that what was done together.
7.Sustainability J. is published in Europe and then it’s more logical to choose the English version of the UK. In such a case, L84 favour, L 90,94,99,103,111,119,158,159,162 analysed; L261 Centre.
8.Authors from Number 9 References switched to describing article names in capital letters. This must be fixed in lowercase.
9.Supplementary.
The non-significant (p > 0.05) effects of earthworms on soils. Is this a reliable conclusion from the results of the experiment, or is all this easily explained by small replicates (n = 3) ?
++++++++++++++
Specific Comments:
L 3 Eisenia Fetida. Compare L 366, 367. Why other authors Foetida?
L 23 Did the experiment have several soils?
L 26 decomposition; WHAT?
L 27 agricultural soil - Abstract does not contain this mention. See L49
L 54 it is better to replace types with species
L 54 comma: two earthworm types on soil C and N dynamics,
L 56 Where does this hypothesis come from? (M. hilgendorfi have more potential to immobilize C to soils). You would think that she appeared after this study!
L 59 What is the soil?
L 62 seconds add
L 63 What is the granulometric composition of the soil? (See importance on L 217-218). Why are such small quantities bulk density was 0.48 ± 0.03 g cm−3? This does not happen in the mineral horizon of the soil.
L 64 Method for determination of total carbon and nitrogen
L 65 kg−1
L 70 It's right? Maybe glass?
L 84 Specify a record:25 mg of dry matter g−1 fresh wt day−1 as…
L 86 Here you need to enter an abbreviation for the subsequent: (М) (Е)
L 118 0.01M HCl) was added,
L 121 sulphate.
L 135 (to pass) passing
L 142 mins = …..
L 149 2.5. Measurements of soil respiration
L 164 P-value
L 167 pH in water or salt extract?
L 168 M. hilgendorfi
L 200 (Also,) In addition,
L 209 editorial changes: decreased MBC Aira et al. [37].
L 247-249 M. hilgendorfi E. fetida
L 296 CO2
L 304 specify Amended Rice? Wheat
L 325 specify Rej?ek
L 343 fix Fer.t
L 349 specify humus? Enzyme
L 363 Lumbricus Rubellus

Round 2
Reviewer 2 Report
The authors carefully finalized the article according to the comments of the reviewer in the first round and now the work can be published.
Author Response
Thank you very much. We appreciate for all the detailed and appropriate comments to improve this manuscript.